# Implementing a digital solution for patients with migraine—Developing a methodology for comparing digitally delivered treatment with conventional treatment: A study protocol

Triinu Niiberg-Pikksööt[1,2,3]☯*, Kariina Laas[4]☯, Anu Aluoja[5,6]☯, Mark Braschinsky[2,3,7]☯

1 Neurosciences, Institute of Clinical Medicine, Faculty of Medicine, University of Tartu, Tartu, Estonia, 2 Headache Clinic, Department of Neurology, Tartu University Hospital, Tartu, Estonia, 3 Migrevention OÜ, Tallinn, Estonia, 4 Institute of Psychology, University of Tartu, Tartu, Estonia, 5 Department of Psychiatry, Institute of Clinical Medicine, Faculty of Medicine, University of Tartu, Tartu, Estonia, 6 Psychiatry Clinic, Tartu University Hospital, Tartu, Estonia, 7 Neurology Clinic, University of Tartu, Tartu, Estonia

☯ These authors contributed equally to this work.
* triinu.niiberg-pikksoot@kliinikum.ee

**Data Availability Statement:** All relevant data are within the manuscript. Data collection is currently still in progress, as it is a study protocol. No

## Abstract

Migraine is one of the most frequent and expensive neurological disease in the world. Non-pharmacological and digitally administered treatment options have long been used in the treatment of chronic pain and mental illness. Digital solutions increase the patients' possibilities of receiving evidence-based treatment even when conventional treatment options are limited. The main goal of the study is to assess the efficacy of interdisciplinary digital interventions compared to conventional treatment. The maximum number of participants in this multi-centre, open-label, prospective, randomized study is 600, divided into eight treatment groups. The participants will take part in either a conventional or a digital intervention, performing various tests and interdisciplinary tasks. The primary outcome is expected to be a reduction in the number of headache days. We also undertake to measure various other headache-related burdens as a secondary outcome. The sample size, digital interventions not conducted via video calls, the lack of human connection, limited intervention program, and the conducting of studies only in digitally sophisticated countries are all significant limitations. However, we believe that digitally mediated treatment options are at least as effective as traditional treatment options while also allowing for a significantly higher patient throughput. The future of chronic disease treatment is remote monitoring and high-quality digitally mediated interventions.The study is approved by the Ethics Committee of the University of Tartu for Human Research (Permission No. 315T-17, 10.08.2020) and is registered at ClinicalTrials.gov: NTC05458817 (14.07.2022).

## Author summary

Migraine is a prevalent and costly neurological disease, the treatment of which requires the cooperation of several specialists and the patient self-help. Modern and best treatment

datasets were generated or analyzed during the current study. All relevant data from this study will be made available upon study completion.

**Funding:** This work was supported by Tartu University Clinic Development Fund (MB and TNP). The funders had no role in study design, data collection and analysis, decision to publish, or preparation of the manuscript. URL to sponsors websites: https://www.kliinikum.ee/en/research-and-teaching/clinical-research-centre/.

**Competing interests:** I have read the journal's policy and the authors of this manuscript have the following competing interests: Authors Triinu Niiberg-Pikksööt and Mark Braschinsky are founders of Migrevention OÜ, which develops Migrevention Clinical mobile application and Migrevention Clinical specialist dashboard. Author Anu Aluoja is participated as a consultant in creating CBT module in Migrevention Clinical mobile application. Author Kariina Laas has no competing interests to declare.

consists of several components—medications, patient self-help, nurse counselling, physiotherapy and cognitive-behavioural therapy. To our knowledge, there is a shortage of headache specialists considering the treatment needs. Therefore, we are investigating the possibilities of whether the digitalization of the entire patient treatment journey saves the working time of specialists and allows us to reach more patients. The biggest strength of the study is that to our best knowledge it is a unique randomised controlled study of a novel standardised manual-based intervention for headaches. Among other things, our study focuses on monitoring compliance and compares fully digital intervention with conventional interdisciplinary non-pharmacological treatment. We believe that remote monitoring and high-quality digitally mediated interventions may be the future of many chronic disease management.

## 1. Background

The prevalence of migraine in the adult population of the world is around 12% [1] in Europe 14.7% [2] and in Estonia 17.7% [3]. Migraine is an expensive disease, costing the European economy 95 billion euros annually and migraine sufferers miss between 2 and 46 days of work annually (absenteeism). Presenteeism, which reduces workability, productivity, and efficiency, is high in migraine sufferers and, thereby harms the economy [1,4,5].

In addition to pharmacological treatments, non-pharmacological interventions (NI) are important in migraine management, encompassing patient participation, self-help skills, and various approaches such as patient education, headache nurse counselling, cognitive behavioural therapy and physiotherapy [6–11]. Headache nurses, through education and counselling, have demonstrated positive effects on the functioning of migraine patients, highlighted also in Kristi Tamela's MSc thesis in Estonia [8–10]

While relaxation techniques, progressive muscle relaxation, biofeedback, and cognitive behavioral therapy (CBT) were formerly considered Level A therapies with an excellent safety profile, the 2019 Cochrane systematic review revealed insufficient high-quality data to determine the effectiveness of psychological therapies for migraine prevention and treatment in adults [6,7,11].

The efficacy of physiotherapy, another NI component, varies depending on the headache diagnosis, with contradictory results regarding its effectiveness in migraine treatment. Physiotherapeutic treatment aims to address peripheral and central hypersensitivity, alleviate neck-shoulder strain, and reduce pain and vestibular symptoms [12,13]. Although NI are a promising addition to medication, these interventions require more resources and are not universally accessible to migraine patients.

In the contemporary digital era, technology-based treatments play a pivotal role in monitoring health, organizing interventions, facilitating communication among healthcare providers, and supporting individuals with chronic diseases and mental health issues [14–16]. Digital solutions offer substantial potential to enhance access to education, counselling, and NI, particularly benefiting patients constrained by time, finances, or transportation access [17–27].

Despite existing digital applications for migraine treatment, there is a need for improvement. Predominantly researches are focusing on the applicability of digital diaries and limited is exploration of comprehensive interdisciplinary digital NI and their effectiveness [28].

One of the most popular mobile applications Migraine Buddy records headache days, tracks headache triggers, patterns, coping tactics and more [29]. Similar digital headache diaries (DHD) and assistants can monitor migraines and train health-related habits with minimal

data loss, over 80% compliance, user-friendly solutions, patient acceptance, and a low perceived burden on daily life [30].

Digital CBT and digitally administered relaxation techniques has been shown to be successful in the treatment of mood and anxiety disorders, insomnia, and can be used to ease other accompanying emotional difficulties in people with migraine [31–36]. Digital physiotherapy, cost effective and accessible through online programs, aligns with current treatment guidelines for various chronic diseases, promoting physical activity and interdisciplinary treatment concepts [37,38].

Limited studies have investigated the effectiveness of patient's interactions with headache nurses, virtual nurses or virtual agents. For example, the Help4Mood platform uses a virtual avatar for depression treatment [39]. Virtual nurses play a role in collecting medical data, providing assistance, motivating patients, and enhancing treatment adherence [40].

Several authors have concluded that with NI, the creation of an effective, novel, and evidence-based digital therapeutic (DTx) platform would improve patient self-management [40] Notably, there is currently a lack of published articles on comprehensive digital intervention programs specifically designed for migraine.

Studies on technology-based interventions (TBI) and apps for chronic disease primarily emphasize applicability, functionality, user experience, and desirable attributes, with less understanding of their effectiveness in achieving long-term behavioral change [17–27].

Key issues with digital solutions include the absence of real-time therapist/patient interaction, challenges in delivering personalized and timely feedback, difficulties in TBI delivery, regional variability in digital solution availability, issues in generating patient intervention reports or summaries, data protection concerns and potential data loss due to technical, internet connection, programme compatibility, software or hardware issues [12,27,40].

Managing headaches involves a multifaceted approach, encompassing pharmacological and NI treatments, preventative methods, lifestyle changes, and self-monitoring [41,42]. Previous research indicates that users often discontinue digital CBT early, facing challenges in maintaining adherence without sufficient clinical contact [43,44]. Studies on internet-based CBT for depression highlight the impact of depression severity on dropout rates, necessitating continuous monitoring [45]. Future research should explore the influence of anxiety and depression on the treatment adherence of a headache patient. Furthermore, it is unknown whether evidence-based digital therapies can successfully be used outside of clinical practice (i.e., self-help with minimal or no clinical intervention) [41,44].

The effectiveness of DHDs compared to conventional 'paper-pencil' headache diaries (PD) remains uncertain, and their impact on treatment adherence is unclear. In conventional diaries, patient often pre-fill fields or mark the conditions later [46–48]. Previous studies indicate that factors such as age, gender, season, day of the week, headache frequency, headache severity, headache medication use, and DHD completion time may all influence a adherence to diary filling [47]. Identifying the elements, background factors and reminders influencing DHD completion is crucial for this study.

Future studies on digital interventions should focus on quantifying the impact of digitally mediated behavioral interventions on pain intensity and psychiatric symptoms. No research has determined the 'optimal dose' of behavioral interventions, including the required number and duration of CBT sessions [44]. The effectiveness and successful digital application of evidence-based CBT with minimal clinical involvement also require investigation.

Limited studies have explored virtually mediated nurse counselling and digital physiotherapy. While a virtual nurse (avatar) may address healthcare challenges, it cannot diagnose or collaborate with other team members [40]. Online chatroom counselling with nurses offers opportunities to enhance clinical contact in digitally mediated interventions, but various limitations and challenges need addressing [49]. To our knowledge, the efficacy of digital

physiotherapy in interdisciplinary headache treatments remains unstudied. A global shortage of headache specialists, including clinical psychologists, CBT therapists, physiotherapists, and headache nurses, contributes to long waiting lists and limited availability. Geographic constraints further impede access to services, particularly for those residing far from large centers. Digital NI options address these challenges by expanding access to services, utilizing professional intervention methods, reducing dropout rates, and thereby improving intervention efficacy [50,51].

The main goal of the present study is to assess the efficacy of interdisciplinary digital interventions compared to conventional treatment options.

The essential secondary objectives encompass evaluating individual and combined components of interventions, including adherence, credibility, clinical contact, and cost-effectiveness in comparison to conventional format.

## 2. Methods and design

### 2.1 Ethics statement

The procedures for and design of the study were approved by the Ethics Committee of the University of Tartu for Human Research (Permission No. 315T-17) and the study is registered at ClinicalTrials.gov: NCT05458817.

### 2.2 Study design

This is a multi-centre, open-label, prospective, randomized, and controlled study on the interdisciplinary intervention of migraine with digital technology-mediated treatment options versus conventional treatment options.

In the preparation phase of the study, migraine patients and specialists from Estonia contributed to the project. They assessed the Migraine Buddy application [29] and found it to be overly difficult, especially during the aura and attack phases of a migraine. The data of the application must also be available to specialists and compatible with e-health.

While preparing and planning the study, it became clear that no interdisciplinary digital solution for the treatment of headaches currently exists. Existing solutions are mostly digital headache diaries with individual appendices. There is no solution that brings comprehensive treatment, including nurse counselling, psychotherapy, and physiotherapy, to the patient.

In 2020, Migrevention started developing an interdisciplinary mobile application and specialist dashboard for migraine patients. They conducted validation interviews with Estonian migraine patients before developing the digital solution. Validation interviews showed that patients are ready to implement digital education, CBT, and physiotherapy programs into their treatment plan. The development of the Migrevention application provided a good opportunity to study a digitally mediated interdisciplinary headache treatment and compare it with conventional treatment options. For this reason, we use the Migrevention application, which monitors the patient's use of the system, andobtains information about patient preferences, their health behavior, the actual application of patients' self-help skills, their need for clinical intervention, etc. The digital format allows for reminders and follow-ups on patients, enabling researchers or clinicians to monitor data in real-time. The importance of this has also been emphasized in previous studies [52]

### 2.3 Planned sample size

In order to devise the sample size, we conducted an estimated power analysis to establish the size of the groups (75 patients in each group) and strive for 80% power. Hence, the maximum estimated number of participants is 600.

The sample comprises consecutive migraine patients who meet the inclusion criteria and visit a neurologist in a partner clinic. In section 2.5, the precise involvement of patients is outlined.

Consent to participate in the research is voluntary and does not affect the patient's ability to use the Migrevention solution or their usual minimal treatment options.

### 2.4 Participants inclusion and exclusion criteria

Inclusion criteria:

1. Diagnosis of frequent episodic migraine, headache days per month 4–14 days

2. Age 18–64 years

3. Very good command of the Estonian language, both orally and in writing

   Exclusion criteria:

1. All other diagnoses of primary and/or secondary headaches

2. Currently existing severe depression

3. History of psychotic disorder(s)

4. Pregnancy or lactation

5. Severe somatic disorder(s)

6. History of severe organic psychiatric disorder(s)

7. History of other chronic pain condition(s)

8. History of addictive disorder(s)

### 2.5 Primary outcome measure

The primary outcome measure is a reduction in the number of headache days (data derived from diaries) three months after the end of the interdisciplinary intervention programmes compared to the baseline (at the start of intervention).

### 2.6 Secondary outcome measures

The secondary outcome measures are as follows: number of headache days six and nine months after finishing intervention (follow-up) (data derived from diaries); change in consumption of analgesics or triptans (data derived from diaries); change in pain intensity (data derived from diaries); changes in depression, anxiety scores (data derived from tests); increase in pain acceptance (data derived from tests); changes in quality of life (data derived from tests); changes in number of sick-leave days (data derived from the national e-health system); and assessment of impact on the burden of migraine through sick-leave days, disability rate, and specialist visits (data derived from e-health systems). Specific methods are presented in Table 1 and Table 2.

### 2.7 Patients inclusion and specialists involvement process

The neurologist examines the eligibility of the patient for the study, explains their opportunity to join the study, and presents an informed consent form. Within three business days, the study team contacts the patient, provides answers to any patient questions, asks patients to

**Table 1. Methods used in the study.**

| Indicator | Method | Time | Procedures for use | Patient time for filling test | Investigator |
|---|---|---|---|---|---|
| Frequency of headaches | Digital diary, paper diary | Continuous | Continuous | > 2 minutes | Patient independently in the application or on paper |
| Medication use | Digital diary, paper diary | Continuous | Continuous | > 2 minutes | Patient independently in the application or on paper |
| Medication use | Prescription Centre | Continuously/ Intervals [1] | Continuously/ Intervals [1] | N/A | Specialist desktop/doctor or nurse. |
| The effect of headaches on everyday life | Headache Impact Test (HIT-6) [53] | Intervals [1] | Intervals [1] | 2–5 minutes | The patient performs independently in the application or the test environment REDCap. |
| Screening for mental disorders | Emotional State Questionnaire (ESQ-2) [54] | Intervals [1] | Intervals [1] | 2–5 minutes | The patient performs independently in the application or the test environment REDCap. |
| Mental disorders | Diagnostic and Statistical Manual of Mental Disorders, Fifth Edition (DSM-V) symptom checklist + Psychiatrist consultation (if DSM-V symptom checklist detects a mental health disorder | Intervals [3] | Intervals [3] | 60 minutes | The patient performs independently in the application or the test environment REDCap and/or one psychiatrist appointment |
| Personality profile | Swedish Universities Scales of Personality (SSP) [55] | Once at the beginning of the study | Once at the beginning of the study | 30 minutes | The patient performs independently in the the test environment REDCap. |
| Pain acceptance | Chronic Pain Acceptance Questionnaire-Revised (CPAQ-R) [56] | Intervals [3] | Intervals [3] | 2–5 minutes | The patient performs independently in the application or the test environment REDCap. |
| Pain-related anxiety | Pain Anxiety Symptoms Scale (PASS) [57] | Intervals [3] | Intervals [3] | 2–5 minutes | The patient performs independently in the application or the test environment REDCap. |
| Quality of life | EUROHIS-QOL 8-item index [58] | Intervals [2] | Intervals [2] | 2–5 minutes | The patient performs independently in the application or the test environment REDCap. |
| Patient satisfaction with treatment | Headache Attributed Lost Time (HALT) [59, 60], Headache Under-Response to Treatment (HURT) [59] | Intervals [3] | Intervals [3] | 2–5 minutes | The patient performs independently in the application or the test environment REDCap. |
| Patient satisfaction with digital therapy | Satisfaction questionnaire | Intervals [3] | Intervals [3] | >2 minutes | The patient performs independently in the application or test environment REDCap. |
| Specialist satisfaction with digital treatment methods | Satisfaction questionnaire | Intervals [3] | Intervals [3] | > 2 minutes | The specialist performs independently in the test environment REDCap. |
| Cost-effectiveness | Queries from the statistical databases of the Estonian Health Insurance Fund, the Social Insurance Board, and the Institute for Health Development | Intervals [2] | Intervals [2] | N/A | Health economist |
| Patient's willingness to pay for headache treatment | Willingness to pay bidding game | Intervals [2] | Intervals [2] | 2–5 minutes | The patient performs independently in the application or the test environment REDCap. |
| Demographics | Demographic Questionnaire | At the beginning of the study | Once at the beginning of the study | > 2 minutes | The patient performs independently in the application or the test environment REDCap. |

[1] At intervals—at the beginning of the study, at 8-week intervals, at the end of the study

[2] At intervals—at the beginning of the study, at the end of the study

[3] At intervals—at the beginning of the study, at the end of the study, in a follow-up study

**Table 2. The data collected by testing in all groups.**

| Method | Collected data |
| --- | --- |
| Participant demographics | Gender, age, place of residence, nationality, education, marital status, employment, income |
| Emotional State Questionnaire (EST-Q) test results | Subscales for depression, generalised anxiety disorder, panic disorder, insomnia, and asthenia |
| Diagnostic and Statistical Manual of Mental Disorders, Fifth Edition (DSM-V) symptom checklist results | Depression, anger, mania, anxiety, somatic symptoms, suicidal ideation, psychotic symptoms, sleep problems, memory problems, compulsive thoughts and behaviors, dissociation, personality dysfunction, and drug use |
| EUROHIS QOL-8 | Quality of life, satisfaction with health, satisfaction with the ability to do everyday life activities, satisfaction with oneself, satisfaction with personal relationships, satisfaction with living conditions, is there enough energy for daily activities, is there enough money to meet one's needs |
| Headache Impact Test (HIT-6) | Effect of headache on patient life |
| Headache Attributed Lost Time (HALT) | Loss of time due to headache |
| Headache Under-Response to Treatment (HURT) | Response to treatment |
| Pain Anxiety Symptoms Scale (PASS) | Pain-avoiding behavior, frightening thoughts, cognitive anxiety, physiological reactions to anxiety |
| Chronic Pain Acceptance Questionnaire-Revised (CPAQ-R) | Participation in activities, pain tolerance, pain acceptance |
| Swedish Universities Scales of Personality (SSP) | Somatic trait anxiety, psychic trait anxiety, stress susceptibility, lack of assertiveness, impulsiveness, adventure-seeking, detachment, social desirability, embitterment, trait irritability, mistrust, verbal trait aggression, physical trait aggression |
| Willingness to pay questions (WTP) | Willingness to pay for headache treatment |
| Patient Satisfaction Questionnaire | Satisfaction with mobile application; Satisfaction with mobile application design (UI); Satisfaction with patient mobile application user experience (UX); Satisfaction with patient mobile application structure; Satisfaction with patient mobile application possibilities; information about the extent to which the mobile application facilitated the subject's communication with the specialist; satisfaction with digital treatment compared to conventional treatment; whether digital treatment interventions could completely replace conventional treatment; whether the patient wants to continue using the patient application; what suggestions are made to improve the application; what other features patients want to see in the mobile application |
| Specialist Satisfaction Questionnaire | Satisfaction with the specialist desktop; whether the desktop makes work easier; satisfaction with the specialist desktop design (UI); satisfaction with specialist desktop user experience (UX); satisfaction with the logic of specialist desktop; satisfaction with specialist desktop possibilities; whether the desktop facilitates communication with patients; whether the digital intervention has improved patient satisfaction with the service; what suggestions are made to make the desktop better; what other features you want to see in the mobile app |

sign the informed consent form digitally, and conducts a randomization. Subjects are divided among eight groups according to randomization, and the treatment intervention continues according to the manual.

For digitally administered intervention, a separate desktop (Migrevention Clinical specialist dashboard) will be developed for the specialists, which is accessible via the work computer of the specialist and is cross-linked to the mobile application of the patient (Migrevention

Clinical). Specialists who treat participants via the Migrevention digital solution are assigned a separate working time and a so-called remote reception time.

The usual clinical face-to-face contact during conventional treatment intervention occurs as a regular consultation according to a detailed intervention plan.

Clinical face-to-face contact during a digital treatment intervention occurs through video consultation if a specialist recognizes the urgent need for it (for example, the depressive and/or anxiety scores of the patient increase).

## 2.8 Settings and randomization process

The study will be conducted in parallel for each intervention setting—the four study groups (digital treatment groups) and the four control groups (standard interdisciplinary treatment groups or treatment standard groups) (Table 3).

After the initial informed consent, patients who qualify for the study will be added to the research group and assigned a participation number based on the precise time the study group acquired the contact information of the patient (date and time, if necessary). After the informed consent is signed, participants are randomized into eight groups. A randomization table has been created for this purpose, in which 600 distinct numbers are randomly separated into eight groups. The patient is assigned to a group in accordance with the randomization table.

All groups receive the usual minimal treatment consisting of medications in addition to the usual appointment with a neurologist. Control groups receive the usual treatment consisting of medications, the usual appointment with a neurologist, headache nurse, CBT therapist and a physiotherapist, in different combinations. All 'as usual' treatments involve clinical face-to-face contact.

All participants of the study groups receive digital treatment consisting of medications and a neurologist appointment as usual. Additionally, each participant is assigned digital appointments with the following specialists: headache nurse, CBT therapist and/or physiotherapist. These appointments are assigned in different combinations depending on the randomization. The digital interventions are delivered through the Migrevention Clinical solution.

**Table 3. Division of groups.**

| Study group I | Control group I | Study group II | Control group II | Study group III | Control group III | Study group IV | Control group IV |
|---|---|---|---|---|---|---|---|
| UMT[1] | UMT | UMT | UMT | UMT | UMT | UMT | UMT |
| DHD[2] | PD[3] | DHD | PD | DHD | PD | DHD | PD |
| | | DHN[4] | UHN[5] | DHN | UHN | DHN | UHN |
| | | | | dCBT[6] | uCBT[7] | dCBT | uCBT |
| | | | | | | dPT | uPT |

[1]UMT–Usual Minimal Treatment

[2]DHD–Digital Headache Diary

[3]PD–Paper Diary

[4]DHN–Digital Treatment by Headache Nurse

[5]UHN–Usual Treatment by Headache Nurse

[6]dCBT–Digital CBT Programme

[7]uCBT–Usual CBT Programme

[8]dPT—Digital Physiotherapy

[9]uPT–Usual Physiotherapy

**Table 4. Interventions by group.**

| | Study group (digitally mediated NI) | Control group (conventionally mediated NI) |
|---|---|---|
| Headache diary | Participant is filling DHD in the Migrevention Clinical mobile application. | Participant is filling PHD, usually used in conventional treatment. |
| Headache nurse counselling | The headache nurse sends self-help materials (previously recorded audio) to the participant through the Migrevention Clinical specialist dashboard and communicates with the patient through the chatroom connected to the Migrevention Clinical mobile application. The Participant receives notifications to the Migrevention Clinical mobile application about new tasks, listens to the materials, and if needed, asks the headache nurse additional questions via the chatroom. Communication through the chatroom is not limited. | The headache nurse arranges face-to-face meetings with the participant to provide conventional nurse counselling. Meetings take place only at agreed times. |
| Cognitive-behavioral therapy | The clinical psychologist sends CBT based self-help materials (previously recorded audio), diaries (previously built-in) and exercises (previously recorded audio), through the Migrevention Clinical specialist dashboard. The Participant listens to audio materials and fulfils diaries inside the Migrevention Clinical mobile application. The clinical psychologist gives written feedback for the participant after they have filled their diaries. A possibility to give feedback to the patient is built-in to the Migrevention Clinical specialist dashboard. It is also possible for the participant to start a chatroom with the clinical psychologist if needed. Communication through the chatroom is not limited. | The clinical psychologist arranges face-to-face meetings with the participant to provide conventional CBT. Meetings take place only at agreed times. |
| Physiotherapy | The physiotherapist sends self-help materials (previously recorded audio) and exercises (previously recorded video) through the Migrevention Clinical specialist dashboard. The Participant listens to audio materials and performs the assigned exercises according to the videos inside the Migrevention Clinical mobile application. The physiotherapist can give written feedback for the participant after their tasks are done. A possibility to give feedback to the patient is built-in to the Migrevention Clinical specialist dashboard. It is also possible for the participant to start a chatroom with the physiotherapist if needed. Communication through the chatroom is not limited. | The physiotherapist arranges face-to-face meetings with the participant to provide conventional physiotherapy. Meetings take place only at agreed times. |

## 2.9 Data collection

Data collection occurs digitally on the mobile application, in the online research environment of the National Centre for Transitional Medicine and Clinical Research—REDCap, and at specialist receptions in parallel. The volume of data collected varies from group to group and depends on the randomization. Participants in the digital group complete the tests on the mobile application and in the REDCap environment. Participants in the control group fill in the tests and the satisfaction questionnaire in the REDCap environment. Specialists fill in the satisfaction questionnaire in the REDCap environment.

In addition to the data from tests and databases, the following data will be collected:

1) frequency of headache days; 2) pain intensity; 3) medications used; 4) performance of tasks sent or set by the headache nurse; 5) performance of the tasks sent or set by a CBT therapist; 6) performance of the tasks sent or set by a physiotherapist.

During the research, several parameters will be measured. The longest time will be devoted to the patient's completion of the tests at the beginning and end of the study—2–2.5h. The time required to complete the tests each week is between 5and 10 minutes. The methods used in the study are presented in more detail in Table 2. The testing data collected in all groups are presented in Table 3.

## 2.10 Interventions

NI conducted with the participants are detailed in Table 4. All materials sent to participants through the Migrevention Clinical solution (study group) or explained in a face-to-face

appointment (control group) are based on the intervention manual and are the same in both groups. The discussion topics and materials are distributed in the order specified in the intervention manual. Since the study attempts to mimic a standard conventional intervention as closely as possible, in the case of conventional treatment (control group), the communication between the specialists and participants is not precisely described in the intervention manual, leaving each specialist free rein over communication.

### 2.11 Statistical analysis

The programs SPSS and R will be used for data analysis. The use of the main generalized linear models for data analysis, such as ANOVA, MANOVA and multinomial linear regression analysis, is planned.

### 2.12 Ethics and approvals

All patients wishing to participate in the study will be given a written informed consent form to read during their first appointment with their neurologist. Patients have the right to refuse to participate in the study, to suspend their participation in the study, and to prohibit the use of their data in the study at any time.

On the digital platform and in the REDCap online environment, the data are collected in a personalized form. After the end of the personal intervention and the data quality control, personal data are separated from survey data. The data will be exported from the research server (server of the National Centre for Transitional Medicine and Clinical Research, Quretec server) to a computer for data processing (Microsoft Office Excel). All data are analyzed in confidence with personal users and only for the purposes of this research under the Personal Data Protection Act [[61]. The data are stored in digital form on the server of the National Centre for Transitional Medicine and Clinical Research (REDCap online environment) and the server of Quretec OÜ (digital platform). For data processing, the data are downloaded to the computer (Microsoft Office Excel) in encrypted form. The code key will be in a folder encrypted by the responsible researcher in the Neurology Department of Tartu University Hospital, and only the responsible researcher has access to it. The planned retention period of the data and the code key is 30 years (until 12/2050), which will allow for longitudinal studies, follow-up analysis, and data quality control of the topic in the future, when necessary.

**Potential risks associated with the study.** In this study, the participants are asked to complete questionnaires, some of which are not routinely used in the clinical assessment of this group of patients, thus placing a somewhat greater time burden on the study participants than on regular patients. No patients will be deprived of their medication during the study, their medication will not be affected in any way, and there will be no additional risks for the participants.

## 3. Discussion

Designed by an interdisciplinary team, our research aims to compare digitally mediated NI migraine treatment options with conventional ones, determine the optimal approach for enhancing patient health and quality of life, optimizing specialists' time, and extending assistance to more migraine patients.

A notable strength of this study is its status as a prospective randomized controlled study of a novel, standardized, manual-based intervention that focuses on, among other things, monitoring compliance and compares fully digital intervention with conventional interdisciplinary NI. Another strength of the study is the assessment of the efficacy and applicability of each treatment component separately, in addition to the whole treatment protocol.

The authors hope that the study helps to overcome the challenges in the field of digital therapeutics (DTx) [62]. The study provides an opportunity to begin to determine the effectiveness of DTx in the treatment of migraine. The implementation of the project also contributes to the integration of DHD data with the e-health system so that they are usable for all providers (including general practitioners) as well as the connection of DHDs to electronic health records.

Interdisciplinary intervention in migraine treatment is known to be effective, especially in preventing migraine chronification [63]. Given the high risk of migraine chronification, long-term treatment adherence and follow-up may be essential to ensure the long-term effectiveness of digital interventions [64]. Based on the above, the research plans to conduct long-term follow-up studies (3, 6, 9 months after the end of the intervention).

There are some limitations that may affect the conduct and results of the study and must be considered when interpreting the results in the future.

The initial sample size will see at least 75 patients in each group, according to the Power analysis. Due to the time-consuming conventional interventions and the time needed to collect data, the realistic obtainable group size within the given timeframes of the study may be smaller. The study may favour the larger clinics and research centers for adopting the protocol and conducting the study and limit similar possibilities of smaller centers. In order to overcome this problem, interim analyses can be undertaken to examine statistically significant associations in a smaller number of patients.

Digital interventions, which are not conducted via video call, are unique strategies that may not be the preference of many individuals in certain situations. In digital interventions, the authors have planned patient initiated clinical contact only via chatroom. Studies on whether and how much clinical contact is required are controversial [14, 65]. The lack of human connection may affect patient adherence, increase the drop-out rate and limit treatment efficacy for some patients. The predetermined and limited number of techniques for use by the specialists may affect the flexibility of treatment. It must also be considered that the entire intervention programme is a minimum programme, and minimal intervention may not have sufficient power to prove the effectiveness of NI migraine treatment.

The whole intervention being done on Estonian patients is a limitation by itself. Estonia is a digitally advanced country whose residents use digital solutions daily. These settings may not be equivalent in other countries and cultures. Hence, the developed methodology is best applicable in countries with similar settings and consequently must be repeated in other countries with different settings.

The authors intend to demonstrate that digitally mediated treatment is at least as effective as conventional treatment but allows for a significantly higher patient throughput, saves/improves specialist workload and is cost-effective. The authors expect remote monitoring and high-quality digitally mediated interventions to be the future of chronic disease treatment.

## Supporting information

**S1 Text. List of abbreviations.**
(DOCX)

## Author Contributions

**Conceptualization:** Triinu Niiberg-Pikksööt, Kariina Laas, Anu Aluoja, Mark Braschinsky.

**Data curation:** Triinu Niiberg-Pikksööt.

**Formal analysis:** Triinu Niiberg-Pikksööt, Mark Braschinsky.

**Funding acquisition:** Mark Braschinsky.

**Investigation:** Triinu Niiberg-Pikksööt.

**Methodology:** Triinu Niiberg-Pikksööt, Kariina Laas, Anu Aluoja, Mark Braschinsky.

**Project administration:** Triinu Niiberg-Pikksööt.

**Resources:** Triinu Niiberg-Pikksööt, Mark Braschinsky.

**Software:** Triinu Niiberg-Pikksööt, Mark Braschinsky.

**Supervision:** Kariina Laas, Anu Aluoja, Mark Braschinsky.

**Validation:** Triinu Niiberg-Pikksööt, Kariina Laas, Anu Aluoja, Mark Braschinsky.

**Visualization:** Triinu Niiberg-Pikksööt.

**Writing – original draft:** Triinu Niiberg-Pikksööt.

**Writing – review & editing:** Kariina Laas, Anu Aluoja, Mark Braschinsky.

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
