## [Decision Letter · Decision Letter 0]

29 Aug 2023

PDIG-D-23-00220

Implementing a digital solution for patients with migraine - developing a methodology for comparing digitally delivered treatment to conventional treatment: A study protocol.

PLOS Digital Health

Dear Dr. Niiberg-Pikksööt,

Thank you for submitting your manuscript to PLOS Digital Health. After careful consideration, we feel that it has merit but does not fully meet PLOS Digital Health's publication criteria as it currently stands. Therefore, we invite you to submit a revised version of the manuscript that addresses the points raised during the review process.

Please submit your revised manuscript within 60 days Oct 28 2023 11:59PM. If you will need more time than this to complete your revisions, please reply to this message or contact the journal office at digitalhealth@plos.org. Please include the following items when submitting your revised manuscript:

We look forward to receiving your revised manuscript.

Kind regards,

Haleh Ayatollahi

Section Editor

PLOS Digital Health

Journal Requirements:

2. We ask that a manuscript source file is provided at Revision. Please upload your manuscript file as a .doc, .docx, .rtf or .tex.

3. Please amend your Data Availability Statement and indicate where the data may be found. No Supporting Files was uploaded in File Inventory.

Additional Editor Comments (if provided):

Reviewers' comments:

Reviewer's Responses to Questions

**Comments to the Author**

1. Does this manuscript meet PLOS Digital Health’s publication criteria? Is the manuscript technically sound, and do the data support the conclusions? The manuscript must describe methodologically and ethically rigorous research with conclusions that are appropriately drawn based on the data presented.

Reviewer #1: Partly

Reviewer #2: Partly

Reviewer #3: Partly

Reviewer #4: Yes

2. Has the statistical analysis been performed appropriately and rigorously?

Reviewer #1: No

Reviewer #2: N/A

Reviewer #3: N/A

Reviewer #4: Yes

3. Have the authors made all data underlying the findings in their manuscript fully available (please refer to the Data Availability Statement at the start of the manuscript PDF file)?

Reviewer #1: No

Reviewer #2: No

Reviewer #3: Yes

Reviewer #4: Yes

4. Is the manuscript presented in an intelligible fashion and written in standard English?

Reviewer #1: No

Reviewer #2: Yes

Reviewer #3: No

Reviewer #4: Yes

5. Review Comments to the Author

Reviewer #1: It is a protocol to compare digitally derived treatment and convectional treatment in migraine.

It is an interesting protocol. Unfortunately, the manuscript has many generalizations that may apply only to some cases and the authors tacitly mention them as applicable to all cases. It is important that authors be more careful and correct such generalizations for the sake of the students reading the journal, so that they do not get the idea that the comments apply to all cases.

Examples:

Lines 50-52: “…patients access to treatment options is limited because there are few specialists worldwide”. Is that truth? How many neurologists, general physicians, pain specialists, anesthesiologists dedicated to treat pain, nurses that assisting in pain management are worldwide?

Lines 57-58: “…high quality digitally mediated interventions are the future of chronic disease treatment”, is that truth in all chronic diseases?

Lines 62-53: ”…. Migraine is one of the most expensive diseases in the world”. How many are the most expensive diseases in the world and what number does migraine occupy?

The manuscript results confusing due to the lightness with which it was written.

In lines 135 – 140, the authors explain that it is difficult to get patient adherence to CBT it is delivered digitally because there is no contact or it is minimal, or because there is anxiety and depression. In the protocol the mention that one of the exclusion criteria is depression, but why they support digitally delivered CBT if they are saying in lines 135-140 that one of the no-adherence reasons is that there is no contact or it contact is minimal.

It would be good to re-write the manuscript with more precision on the statements.

It is necessary to explain with more detail the statistical analysis.

The answer of question 3 is no because it is a protocol, not a study with results.

Reviewer #2: I have reviewed the protocol entitled "Implementing a digital solution for patients with migraine - developing a methodology for comparing digitally delivered treatment to conventional treatment: A study protocol". The Authors designed a study to address the clinical effectiveness of digital interdisciplinary intervention using a Migraine Buddy© application.

The study protocol is technically sound and address the main points of a study protocols.

I have some minor concerns about the study:

1- Statistical analysis should include statistical methods for causal effect inference (e.g. targeted maximum likelihood extimation). This is important to evaluate the additive effect of each arm of the study.

Reviewer #3: This is a study protocol for a randomized controlled trial evaluating four digital interventions (of increasing comprehensiveness) for migraine, compared to standard care. The main content of the manuscript was sound; my main suggestions involve making the manuscript clearer, as it was hard to follow and understand. See my comments below: 

Major comments:

1. Please proofread the whole article again, as there are various grammatical errors or typos throughout. 

2. It was unclear from the introduction that the study will have four conditions (each with increasing numbers of intervention components) and four control groups - it sounded like there would be one comprehensive/multi-component intervention and one control group. 

3. The study's results will contribute to an important gap in the literature (few studies on *comprehensive* digital interventions for migraine, as well as few studies on the “dose” of each intervention component necessary), but this is not explicitly stated or highlighted as much as it should, leaving the reader confused as to what this study would specifically contribute to the literature. The authors should state/highlight this point more in both the introduction and the discussion sections. 

4. If there are four experimental conditions and four control groups, does that mean there will be four pair-wise comparisons for each outcome at each time point? I would suggest adding how you will adjust for multiple comparisons. 

Minor comments:

1. Make the language clearer that the trial has not been run yet (i.e., use more future tense rather than present tense) throughout the manuscript; as the reader, it was hard to discern whether you were talking about events that had already happened, were happening presently, or will happen in the future. 

2. Lines 34-35: Clarify the language here as it is unclear whether you mean data from this particular study or another study. 

3. Lines 151-156: It is unclear whether this paragraph is about this study in particular or future directions (not including this study). 

4. Move section 2.1 to the introduction. 

5. The paragraph starting in 175 seems out of place. I suggest replacing this paragraph with a paragraph highlighting how this study adds to the literature - there have been few (if any) previous studies on comprehensive digital interventions, and it is also important to isolate the effective components of a comprehensive intervention. 

6. In lines 391-395, the authors mention a small-scale pilot study. It doesn’t seem like that study a part of this manuscript (rather, it is a separate study), but the result (that specialist time is reduced more than tenfold) is mentioned in the abstract. If the pilot study was separate from this manuscript, I suggest removing this result from the abstract. If the pilot study was part of this manuscript, I suggest explaining it and the result in much more detail.

Reviewer #4: The subject of this article is an important one and should be further explored in future research. The paper presents a stepping stone for the creation of frameworks in which to assess the digital delivery of migraine treatment. Although the text may need further proofreading work to correct minor grammatical errors, the English language used does not hinder the reader from comprehending the overall message of the article. I commend the authors for tackling such a relevant subject especially in the age of telehealth, telemedicine, and digital health. A series of articles concerning topics around migraine warrants further investigation and publication to provide knowledge for the general public, as well.

6. PLOS authors have the option to publish the peer review history of their article (what does this mean?). If published, this will include your full peer review and any attached files.

**Do you want your identity to be public for this peer review?** For information about this choice, including consent withdrawal, please see our Privacy Policy.

Reviewer #1: Yes: Cleva Villanueva

Reviewer #2: No

Reviewer #3: No

Reviewer #4: No

---

## [Decision Letter · Decision Letter 1]

14 Dec 2023

PDIG-D-23-00220R1

Implementing a digital solution for patients with migraine - developing a methodology for comparing digitally delivered treatment with conventional treatment: A study protocol.

PLOS Digital Health

Dear Dr. Niiberg-Pikksööt,

Thank you for submitting your manuscript to PLOS Digital Health. After careful consideration, we feel that it has merit but does not fully meet PLOS Digital Health's publication criteria as it currently stands. Therefore, we invite you to submit a revised version of the manuscript that addresses the points raised during the review process.

Please submit your revised manuscript within 30 days Jan 13 2024 11:59PM. If you will need more time than this to complete your revisions, please reply to this message or contact the journal office at digitalhealth@plos.org. Please include the following items when submitting your revised manuscript:

We look forward to receiving your revised manuscript.

Kind regards,

Haleh Ayatollahi

Section Editor

PLOS Digital Health

Journal Requirements:

Additional Editor Comments (if provided):

I appreciate the authors for their time and efforts to revise the manuscript. Please address the following minor issues in your next revision, too.

1- Please add appropriate keywords using the MeSH terms.

2- Please remove the subheadings from the introduction section and provide an integrated introduction section. Moreover, the introduction section needs to be summarized in 2-3 pages.

3-Overall, the manuscript is very long and I suggest to move tables to the Appendix section, and summerize different sections of the manuscript.

4- The discussion section needs to be supported by more references.

Reviewers' comments:

Reviewer's Responses to Questions

**Comments to the Author**

1. If the authors have adequately addressed your comments raised in a previous round of review and you feel that this manuscript is now acceptable for publication, you may indicate that here to bypass the “Comments to the Author” section, enter your conflict of interest statement in the “Confidential to Editor” section, and submit your "Accept" recommendation.

Reviewer #1: All comments have been addressed

Reviewer #2: All comments have been addressed

Reviewer #3: All comments have been addressed

2. Does this manuscript meet PLOS Digital Health’s publication criteria? Is the manuscript technically sound, and do the data support the conclusions? The manuscript must describe methodologically and ethically rigorous research with conclusions that are appropriately drawn based on the data presented.

Reviewer #1: Yes

Reviewer #2: Yes

Reviewer #3: Yes

3. Has the statistical analysis been performed appropriately and rigorously?

Reviewer #1: Yes

Reviewer #2: Yes

Reviewer #3: N/A

4. Have the authors made all data underlying the findings in their manuscript fully available (please refer to the Data Availability Statement at the start of the manuscript PDF file)?

Reviewer #1: Yes

Reviewer #2: Yes

Reviewer #3: Yes

5. Is the manuscript presented in an intelligible fashion and written in standard English?

Reviewer #1: Yes

Reviewer #2: Yes

Reviewer #3: Yes

6. Review Comments to the Author

Reviewer #1: The authors addressed all the comments of the reviewers and answered the questions accordingly. The manuscript was modified and is almost ready to be accepted. It is necessary to write the references uniformly, some have date, other do not have dates.

Reviewer #2: I have reviewed the revised version of the paper entitled "Implementing a digital solution for patients with migraine - developing a methodology for comparing digitally delivered treatment with conventional treatment: A study protocol".

The Authors addressed all my comments.

Reviewer #3: I thank the authors for taking my previous suggestions into consideration and improving the manuscript. 

My last suggestion is for the authors to reread and consider restructuring the discussion section -- lines 398-399 seemed out of place, as did lines 393-397 mentioning the pilot study, which was not the study in question.

7. PLOS authors have the option to publish the peer review history of their article (what does this mean?). If published, this will include your full peer review and any attached files.

**Do you want your identity to be public for this peer review?** For information about this choice, including consent withdrawal, please see our Privacy Policy. 

Reviewer #1: No

Reviewer #2: No

Reviewer #3: No

---

## [Editor Report · Decision Letter 2]

18 Jan 2024

Implementing a digital solution for patients with migraine - developing a methodology for comparing digitally delivered treatment with conventional treatment: A study protocol.

PDIG-D-23-00220R2

Dear Mrs Niiberg-Pikksööt,

We are pleased to inform you that your manuscript 'Implementing a digital solution for patients with migraine - developing a methodology for comparing digitally delivered treatment with conventional treatment: A study protocol.' has been provisionally accepted for publication in PLOS Digital Health.

Best regards,

Haleh Ayatollahi

Section Editor

PLOS Digital Health